# Active Teacher Selection for Reinforcement Learning from Human Feedback

## Abstract

Reinforcement learning from human feedback (RLHF) enables machine learning systems to learn objectives from human feedback. A core limitation of these systems is their assumption that all feedback comes from a single human teacher, despite querying a range of distinct teachers. We propose the *Hidden Utility Bandit* (HUB) framework to model differences in teacher rationality, expertise, and costliness, formalizing the problem of learning from multiple teachers. We develop a variety of solution algorithms and apply them to two real-world domains: paper recommendation systems and COVID-19 vaccine testing. We find that the *Active Teacher Selection* (ATS) algorithm outperforms baseline algorithms by actively selecting when and which teacher to query. The HUB framework and ATS algorithm demonstrate the importance of leveraging differences between teachers to learn accurate reward models, facilitating future research on active teacher selection for robust reward modeling.

## 1 Introduction

Specifying objective functions for machine learning systems is challenging, and misspecified objectives can be hacked (Pan et al., 2022; Skalse et al., 2022) or incentivise degenerate behavior (Zhuang & Hadfield-Menell, 2020; Thomas & Uminsky, 2020; Krakovna et al., 2020). Techniques such as *reinforcement learning from human feedback* (RLHF) enable ML systems to instead *learn* appropriate objectives from human feedback (Christiano et al., 2017; Lee et al., 2021; Stiennon et al., 2020). These techniques are widely used to finetune large language models (OpenAI, 2023; Anthropic, 2023; Touvron et al., 2023; Google, 2023) and to train reinforcement learning agents to perform complex maneuvers in continuous control environments (Christiano et al., 2017; Lee et al., 2021). However, while RLHF is relied upon to ensure that these systems are safe, helpful, and harmless (Bai et al., 2022), it still faces many limitations and unsolved challenges (Casper et al., 2023).

In particular, RLHF systems typically rely on the assumption that all feedback comes from a single human teacher, despite gathering feedback from a range of teachers with varying levels of rationality and expertise. For example, Stiennon et al. (2020), Bai et al. (2022) and Ouyang et al. (2022) assume that all feedback comes from a single teacher, but find that annotators and researchers actually disagree 23% to 37% of the time. Reward learning has been shown to be highly sensitive to incorrect assumptions about the process that generates feedback (Hong et al., 2022; Freedman et al., 2021; Skalse & Abate, 2022; Milli & Dragan, 2020), so this single-teacher assumption exposes these systems to dangerous failures (Daniels-Koch & Freedman, 2022). Ideally, RLHF systems should consider the differences between each teacher to improve their safety and reliability.

To leverage multiple teachers in RLHF, we introduce a novel problem called a *Hidden Utility Bandit (HUB)*. A HUB is similar to a multi-armed bandit (MAB), in that at each timestep the agent has a consistent set of alternatives (called "arms") and receives utility based on which it chooses ("pulls"). Unlike a MAB, however, the agent observes the arm's output ("item") *but not the associated utility*. Like in RLHF, it must learn the utility function based on comparison feedback, but unlike in RLHF, the agent can choose amongst multiple teachers. Optimal HUB solutions must therefore actively select *which* teachers to query *when* so as to maximize the expected discounted sum of utilities. Figure 1 shows a simple HUB in which the two arms are vending machines, the two teachers are human taste-testers, and the outputs are fruit.

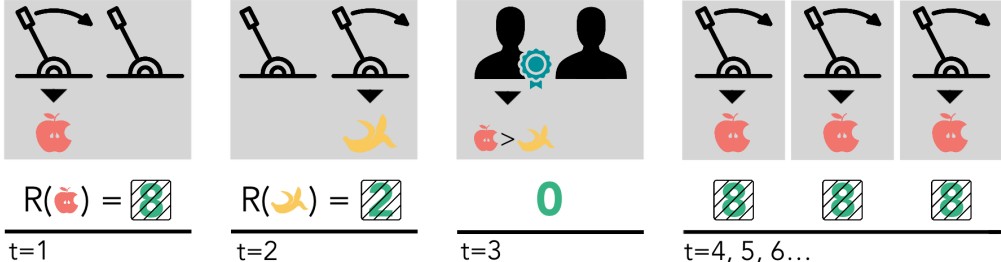

Figure 1: A simple *Hidden Utility Bandit (HUB)* with two arms and two teachers. The agent pulls the first arm, observes an apple, and receives the apple's utility of 8 without observing it. The agent then pulls the second arm, observes a banana, and receives the banana's utility of 2 without observing it. Because these utilities are hidden, the agent foregoes the opportunity for utility on the third timestep to ask the expert teacher which fruit is better. The expert replies that apples are better than bananas, so the agent pulls the first arm to maximize apples for all remaining timesteps.

We present background in Section 2 (and discuss further related work in Appendix A), then formalize the HUB framework and propose a naive baseline solution (Section 3). We then develop an *Active Teacher Selection* (ATS) method that selects *which* teachers to query *when* to maximize cumulative discounted utility (Section 4). Since there are no existing solutions to the novel HUB problem, we introduce multiple families of baseline methods and evaluate these against ATS variations on a realistic paper recommendation task (Section 5.1). ATS outperforms methods with fixed exploration windows, demonstrating the usefulness of selecting *when* to query teachers, and ATS with specific teacher selection outperforms general teacher selection, underscoring the usefulness of selecting *which* teacher to query. As a proof-of-concept, we also demonstrate how this framework can be applied to the real-world problem of evaluating COVID-19 vaccines with noisy tests (Section 5.2). The result is a HUB framework and an ATS algorithm[1] that demonstrate the importance of leveraging differences between teachers to learn accurate reward models. These will facilitate and benchmark improved methods, ultimately leading to scalable reward learning algorithms that learn accurate, robust and value-aligned models.

## 2 BACKGROUND

**Multi-Armed Bandits** *Multi-armed bandits* (MAB) are stateless sequential decision-making problems (Robbins, 1952; Slivkins, 2019). At each timestep $h = 1, 2, \ldots, H$, the agent chooses one of $K$ arms, each with a distribution over utilities. When the agent pulls arm $k \in K$, it receives utility sampled from arm $k$'s distribution $u \sim \mathcal{D}^k$. The agent's goal is to maximize its expected cumulative utility. Our framework is similarly structured, though arm utilities are hidden (as in many real-life applications), and the agent must learn about them from teacher preferences (as in RLHF).

**Partially Observable Markov Decision Processes** *Partially observable Markov decision processes (POMDP)* are sequential decision-making problems where the state of the world is partially hidden from the agent (Littman et al., 1995). A POMDP problem is a tuple $\langle \mathcal{S}, \mathcal{A}, \mathcal{T}, \mathcal{R}, \mathcal{O}, \Omega, \gamma \rangle$, where $\mathcal{S}$ and $\mathcal{A}$ are the state and action spaces, $\mathcal{T}$ and $\mathcal{R}$ are the transition and reward functions, and $\gamma$ is the discount factor. At time $t$, the agent begins in state $s_t$, takes action $a_t$, transitions to state $s_{t+1}$ determined by $\mathcal{T}(s_t, a_t)$ and receives reward $r_t = R(s_t, a_t, s_{t+1})$. However, rather than observing the states directly, the agent observes an observation $\omega_{t+1}$ from the observation space $\mathcal{O}$ determined by the observation function $\Omega(s_{t+1}, a_t)$. A solution to a POMDP is a policy that balances inferring the underlying state and acting in the environment to maximise its expected cumulative reward. While calculating this solution is typically intractable, approximate POMDP algorithms can perform well. Partially observable Monte Carlo planning (POMCP)-style algorithms produce time-efficient online solvers that work by forming a belief tree of fixed depth then using rollouts to estimate the values of the leaf nodes (Silver & Veness, 2010). In this work we use partially observable Monte Carlo planning with observation widening (POMCPOW), a POMCP-style algorithm that uses a weighted particle filter to efficiently produce approximate solutions for problems with large state spaces (Sunberg & Kochenderfer, 2018).

---

[1] Our open-source ATS Julia library is available at `github.com/[redacted]/ATS`.

# 3 HIDDEN UTILITY BANDITS

We design the *Hidden Utility Bandit* (HUB) framework to formalize the problem of reward learning from multiple teachers. Formally, a HUB is a partially-observable sequential decision-making problem consisting of a set of items (each with a distinct utility), a set of arms (each with a fixed distribution over items), and a set of teachers (each with a rationality parameter and cost). We assume that the agent can take one action (pulling an arm or querying a teacher) per timestep. Following an existing standard in work on RLHF (Lee et al., 2021), a HUB models each teacher as Boltzmann-rational with its noisiness modulated by a *rationality parameter* $\beta \in [0, \infty)$. In particular, the probability that a teacher with rationality parameter $\beta$ prefers item $i$ to $j$ is below:

$$\Pr(i \succ j; \beta, \mathcal{U}) = \frac{\exp(\beta \mathcal{U}(i))}{\exp(\beta \mathcal{U}(i)) + \exp(\beta \mathcal{U}(j))}, \tag{1}$$

where $\mathcal{U} : \mathcal{I} \to \mathbb{R}$ gives the true utility of all items in set $\mathcal{I}$.

At each timestep of the HUB problem, the agent chooses between pulling an arm, observing an item sampled from that arm's distribution and receiving *but not observing* that item's utility, or querying a teacher, receiving feedback modulated by that teacher's rationality parameter but incurring that teacher's query cost. We assume that all teachers give feedback based on a single shared utility function. The agent's objective is to maximize the expected discounted sum of utilities, so it must balance querying costly teachers to learn about the utility function with pulling arms to earn utility.

**Definition 3.1.** A *hidden-utility bandit* (HUB) is a tuple $\langle \mathcal{I}, \mathcal{U}, \mathcal{C}, \beta, F, Q, \gamma \rangle$:

- $\mathcal{I}$ is a set of $N$ *items*, each associated with a hidden utility.
- $\mathcal{U} : \mathcal{I} \to [u_{min}, u_{max}]$ is a *utility function* over $\mathcal{I}$, where $\mathbb{U}$ is the utility function space.
- $\mathcal{C} = \{c^1, c^2, \ldots, c^K\}$ is a set of $K$ *arm choices*, each associated with an *arm distribution* $\mathcal{D}^k$ : $\mathcal{I} \to [0, 1]$ giving the probability of returning each item in $\mathcal{I}$, where $\mathbb{D} = \mathbb{D}^1 \times \mathbb{D}^2 \times \cdots \times \mathbb{D}^K$ is the joint arm distribution space over all arm choices $\mathcal{C}$.
- $\beta = \{\beta^1, \beta^2, \ldots, \beta^M\}$ is a set of $M$ *teacher rationality parameters*.
- $F = \{f^1, f^2, \ldots, f^M\}$ is a set of $M$ *teacher query costs*.
- $Q : \mathcal{I} \times \mathcal{I} \to [0, 1]$ is a *query profile* that gives probabilities of picking queries in $\binom{\mathcal{I}}{2}$.
- $\gamma$ is a discount factor.

Here, the agent can observe $\mathcal{I}, \mathcal{C}, \beta, F, Q$, and $\gamma$ but cannot observe the utility function $\mathcal{U}$ or the arm distributions $\mathcal{D}$. At each timestep $t$, the agent can select an arm choice $c_t \in \mathcal{C}$ or a teacher rationality parameter $\beta_t \in \beta$. If the agent pulls an arm choice $c_t \in \mathcal{C}$, it observes an item $i_t$ sampled from the arm distribution $\mathcal{D}^{c_t}$ and receives but does *not* observe the utility $u_t = \mathcal{U}(i_t)$. Conversely, if the agent queries a teacher with rationality parameter $\beta_t \in \beta$, it receives and observes an item pair $(i, j)$ sampled from the query profile $Q$, a preference $p_t$ sampled from *Bernoulli*$(P)$ given the probability $P = \Pr(i \succ j; \beta_t, \mathcal{U})$ in Equation 1, and the teacher query cost $u_t = f^{\beta_t}$.

Since the agent's objective is to maximize the expected discounted sum of utilities $\mathbb{E}[\Sigma_{t=0}^{\infty} \gamma^t u_t]$, it must balance querying teachers to learn about the utility function with selecting bandit arms to earn utility. Standard RLHF systems alternate between fitting a reward model to teacher feedback and learning a policy using the reward model on a predefined schedule. However, the HUB framework allows the agent to interweave these processes to optimize performance.

## 3.1 NAIVE HUB INFERENCE

We propose a naive HUB inference baseline in Algorithm 1. This allows the agent to infer the hidden information: the joint arm distribution $\mathcal{D}^{\mathcal{C}} = (\mathcal{D}^1, \mathcal{D}^2, \ldots, \mathcal{D}^K)$ (common to stochastic multi-armed bandit problems) and utility function $\mathcal{U}$ (unique to the HUB). In Algorithm 1, the agent randomly pulls arms and queries a preselected teacher for a fixed number of timesteps (lines 1-12), approximates the true joint arm distribution and true teacher preference probabilities with sample means (lines 13-14), then uses these to estimate the utility function (lines 15-16). The agent can then simply calculate the the expected utility of each arm and pull the arm with the highest expected utility for the remainder of the episode.

Despite the simplicity of Algorithm 1, it is possible to prove that it converges to the ground truth utility function $\mathcal{U}^*$ and arm distribution set $\mathcal{D}^{\mathcal{C}*}$ in the limit of infinite queries. We prove the following theorem in Appendix B:

---

**Algorithm 1** NAIVEHUBINFERENCE($\cdot$)

---

**Require:** HUB $\langle \mathcal{I}, \mathcal{U}, \mathcal{C}, \beta, F, Q, \gamma \rangle$, $u_{min}$, $u_{max}$, $T$ samples, $\beta^m$ of selected teacher
**Initialize:** *frequency[c], frequency[c][i], frequency[b][q], preferences[b][q]*

---

1: **for** $t = 1, \ldots, T$ **do**
2:     **if** sampleUniformly({TRUE, FALSE}) **then**
3:         sample $c \sim \mathcal{C}$                      ▷ Sample arm uniformly at random
4:         sample $i \sim \mathcal{D}^c$                 ▷ Sample item from (unobserved) arm distribution
5:         *frequency[c]* ← *frequency[c]* + 1
6:         *frequency[c][i]* ← *frequency[c][i]* + 1
7:     **else**
8:         sample $b \sim \beta$                    ▷ Sample teacher uniformly at random
9:         sample $q = (i, j) \sim Q$              ▷ Sample query from query profile
10:        sample $p \sim Bernoulli(\Pr(i \succ j; b, \mathcal{U}))$      ▷ Sample preference given Equation 1
11:        *frequency[b][q]* ← *frequency[b][q]* + 1
12:        *preferences[b][q]* ← *preferences[b][q]* + p
13: $\hat{D}^c(i) \leftarrow \frac{frequency[c][i]}{frequency[c]}$    $\forall c \in \mathcal{C}, i \in \mathcal{I}$           ▷ Estimate arm distributions
14: $\hat{P}(b, q) \leftarrow \frac{preferences[b][q]}{frequency[b][q]}$    $\forall b \in \beta, q \in Q$       ▷ Estimate preference probabilities
15: $\Delta_{ij} = -\frac{1}{\beta^m} \ln\left[\frac{1}{\hat{P}(\beta^m, q=(i,j))} - 1\right]$    $\forall i, j \in \mathcal{I}$    ▷ Calculate using Equation **??**
16: $(x, y) \leftarrow \arg\max_{x,y}[\Delta_{xy}]$             ▷ Find indices of maximum element
17: $\hat{\mathcal{U}}(y) \leftarrow u_{min}$,    $\hat{\mathcal{U}}(i) \leftarrow \left[\frac{u_{max}}{u_{max}-u_{min}}\right] \Delta_{iy} + u_{min}$    $\forall i \in \mathcal{I} \setminus \{y\}$    ▷ Estimate utilities

---

**Theorem 1.** *If the predicted utility function $\hat{\mathcal{U}}$ and the predicted arm distribution $\hat{\mathcal{D}}^C$ are estimated by executing Algorithm 1 with $T$ samples, then $\hat{\mathcal{U}} \rightarrow \mathcal{U}^*$ and $\hat{\mathcal{D}}^C \rightarrow \mathcal{D}^{C*}$ as $T \rightarrow \infty$.*

However, exploring randomly for a fixed number of timesteps and querying a fixed teacher may be suboptimal. By maintaining and updating an internal belief over the hidden information, the agent can query teachers only when teacher feedback is necessary to update its belief.

## 4   ACTIVE TEACHER SELECTION

The *Active Teacher Selection (ATS)* algorithm solves the HUB problem efficiently by maintaining a belief over the utility function and arm distributions, and choosing when to query teachers. This allows it to only query teachers when required for decision-relevant belief updates. ATS can also actively select *which* teacher to query. When teachers are "noisy" ($\beta < \infty$), the preference probability $\Pr(i \succ j; \beta, \mathcal{U})$ correlates with the difference in utility between $i$ and $j$, so it will sometimes be more informative for ATS to select teachers with *lower* $\beta$ values (Michaud et al., 2020; Barnett et al., 2023). Importantly, this removes the need to set the problem-specific hyperparameters in Algorithm 1 for exploration ($T$) and teacher selection ($\beta^m$).

### 4.1   ATS ALGORITHM

The ATS algorithm has two general steps: the HUB is first converted to a simplified partially observable Markov decision process (POMDP) (Littman et al., 1995) and then solved using a Monte Carlo POMDP solver with custom rollout policies.

**Constructing the HUB-POMDP**   The HUB-POMDP state contains the HUB utility function and arm distributions. The HUB-POMDP reward function gives the *expected* utility of each arm according to this state.

**Definition 4.1.** A **hidden utility bandit POMDP (HUB-POMDP)** is a tuple $\langle \mathcal{S}, \mathcal{A}, \mathcal{T}, \mathcal{R}, \Omega, \mathcal{O} \rangle$:

- $\mathcal{S} = \mathbb{U} \times \mathbb{D}$ is the state space: the state $s \in \mathcal{S}$ is a tuple $\langle \mathcal{U}, \mathcal{D}^C \rangle$ that is fixed.
- $\mathcal{A} = \mathcal{C} \cup \beta$ is the action space: the arm choices $\mathcal{C}$ and teachers $\beta$.
- $\mathcal{T} : \mathcal{S} \times \mathcal{A} \rightarrow \mathcal{S}$ is the stationary transition function: $\mathcal{T}(s, a) = s \; \forall_{s \in \mathcal{S}} \; \forall_{a \in \mathcal{A}}$.

- $\mathcal{R} : \mathcal{S} \times \mathcal{A} \to \mathbb{R}$ is the reward function:

$$\mathcal{R}(s,a) = \begin{cases} \Sigma_{i \in \mathcal{I}} \mathcal{U}(i) \mathcal{D}^a(i) & \text{if } a \in \mathcal{C} \\ -f^a & \text{if } a \in \beta \end{cases}$$

- $\Omega : \mathcal{I} \cup \mathbb{P}$ is the observation space: the items $\mathcal{I}$ and query-preferences $\mathbb{P} = \mathcal{I} \times \mathcal{I} \times \{0,1\}$.
- $\mathcal{O} : \mathcal{A} \times \Omega \to [0,1]$ is the observation function:

$$\mathcal{O}(a,\omega) = \begin{cases} \mathcal{D}^a(i) & \text{if } a \in \mathcal{C} \\ Q(i,j) \Pr(i \succ j ; \beta^m = a, \mathcal{U}) & \text{if } a \in \beta \end{cases}$$

Teacher selection can be *general* or *specific*. Under specific selection, the agent chooses which teacher to query. The HUB-POMDP's action space contains all $M$ teachers, $\mathcal{A} = \mathcal{C} \cup \beta$, as shown in the HUB-POMDP above. Under general selection, the agent chooses *when* to query a teacher, but as in RLHF cannot choose *which* teacher to query. The HUB-POMDP's action space is modified to contain a single general teacher selection action, $\mathcal{A} = \mathcal{C} \cup \{\beta^g\}$.

These alternatives offer a tradeoff: general selection reduces the state space size and computational complexity while specific selection provides the agent with additional control over its feedback. Our experimental results (reported in Section 5.1) indicate that specific greatly outperforms general teacher selection, so we will use ATS with specific teacher selection unless otherwise specified.

**Solving the POMDP** While exact POMDP solutions are typically intractable, approximate POMDP algorithms often perform well. *Partially observable Monte Carlo planning (POMCP)* algorithms produce time-efficient online solvers that form a belief tree of fixed depth and use rollouts to estimate leaf node values (Silver & Veness, 2010). *POMCP with observation widening (POMCPOW)* uses a weighted particle filter to efficiently produce approximate solutions for problems with large state spaces (Sunberg & Kochenderfer, 2018), so we adapt it to the HUB-POMDP with specialized rollout policies. We describe and compare candidate rollout policies that we designed specifically for the HUB problem in Appendix D. ATS with the custom *best arm* rollout policy performs best, so we use that POMCPOW vriant for our experiments.

### 4.2 TEACHER NOISE INFERENCE IN ATS

RLHF systems typically assume that the teacher rationality parameters $\beta$ are known. However, as this is sometimes unrealistic, we show in Theorem 2 that $\beta$ can also be estimated from preference data. Specifically, given $\Pr(i \succ j; \beta_m, \mathcal{U})$, it is possible to estimate $\hat{\beta}_m = \frac{1}{z}\beta_m$, where $z$ is a scaling factor determined by $\mathcal{U}$. $z$ is based on the difference $\Delta_{ij} = \mathcal{U}(i) - \mathcal{U}(j)$, so as long as the same comparison pair $(i,j)$ is used, all teacher rationality estimates will be on the same scale. (They can be calculated directly if $\Delta_{ij}$ happens to be known for a specific $(i,j)$.)[2] We prove the theorem below in Appendix C.

**Theorem 2.** *Given two items $i, j \in \mathcal{I}$ where $\mathcal{U}(i) < \mathcal{U}(j)$ and the preference probability $P = \Pr(i \succ j; \beta_m, \mathcal{U})$ from Equation 1 we can estimate $\hat{\beta}_m = \frac{1}{z}\beta_m$ as in Equation 3. If $\Delta_{ij}$ is known, we can further calculate $\beta_m = z \cdot \hat{\beta}_m$, where $z = -\Delta_{ij}^{-1}$.*

$$\hat{\beta}_m = \ln\left(\frac{1}{P} - 1\right). \tag{2}$$

We demonstrate this procedure in our experiments in Section 5.2. In addition, we evaluate this procedure in simulation by setting $\beta = \{0.01, 1.0\}$, running a random policy for 1000 timesteps, estimating $\{\hat{\beta}_1, \hat{\beta}_2\}$, and scaling the estimate so that the greatest value is equal to 1.0. We observe a mean squared error of only 0.061 across 100 simulations, indicating that this procedure is accurate.

## 5 EXPERIMENTS

We apply the HUB framework to two real-world domains: *paper recommendations* and *COVID-19 vaccine testing*. In the recommendation domain, we conduct comprehensive experiments that

---

[2]Note that it is also possible to directly add $\beta$ to the state space of the HUB-POMDP and then solve it, but this increases the size of the state space and makes the problem less tractable.

Figure 2: Paper recommendation as a HUB problem. Paper categories (Application, Benchmark, Theory) are items ($\mathcal{I}$), professors are teachers with rationality ($\beta$) and cost ($F$) parameters, conferences are arms with distributions ($\mathcal{D}$), and relevance scores are utilities ($\mathcal{U}$). The goal is to recommend the most relevant conferences to read papers from.

evaluate the performance of various solution algorithms (Section 5.1), compare rollout simulation policies (Appendix D), and examine the impact of varying teacher query costs (Appendix E). The more complex vaccine domain provides a proof-of-concept, using the HUB framing to address an urgent problem and demonstrating how $\beta$ values can be estimated from real-world data. We find that the HUB framework captures both problems well, that the ATS algorithm outperforms all baselines in comprehensive testing in the recommendation domain, and that ATS is the best-performing algorithm that also identifies the best vaccine in the vaccine domain proof-of-concept.

**Algorithms** We fix ATS to use *specific* teacher selection and the *best arm* rollout policy unless otherwise specified. To our knowledge, the HUB problem is novel and has no solutions in prior literature, so we construct multiple families of baseline methods (*naive* and *random*) for comparison. *Naive* algorithms choose randomly amongst pulling arms and querying the selected teacher for $T$ timesteps, use these observations to estimate the arm distributions and utility function (using Algorithm 1), and then pull the arm with the highest estimated expected utility at each subsequent timestep. Naive algorithms require problem-specific hyperparameters $\beta^m$ and $T$, so for these experiments we select the intermediate of 3 teachers ($\beta^m = \beta^2$) and test a range of exploration horizons ($T \in [50, 100, 200]$). *Random* algorithms select actions uniformly at random from a given set. We evaluate a random algorithm that selects actions from the entire action space, as well as one that selects only arms.

## 5.1 CONFERENCE RECOMMENDATION DOMAIN

In the recommendation domain, the system recommends AI conferences from which to read relevant papers. There are three paper categories (Application, Benchmark, Theory) with specified relevance scores, and three conferences (ICLR, ICML, AAAI) with different paper category compositions[3]. The recommender cannot directly observe the relevance scores, so it must learn them by asking professors, whose judgements vary from completely random ($\beta^1 = 0$) to highly accurate ($\beta^3 = 50$). In these experiments, query costs are always 0. (See Appendix E for experiments varying query costs.) Each day, the system recommends one conference, a paper is sampled from that conference's distribution, and the system earns a hidden utility score representing that paper's category's relevance. Alternatively, the system queries a professor who provides a preference over a pair of paper categories. Applying the HUB framework, paper categories are the item set $\mathcal{I} = \{A, B, T\}$, relevance scores are the hidden utility function $\mathcal{U}$, conferences are arm choices $\mathcal{C} = \{c^1 = \text{ICLR}, c^2 = \text{ICML}, c^3 = \text{AAAI}\}$, and professors are teachers with rationality $\beta = \{\beta^1 = 0, \beta^2 = 0.01, \beta^3 = 50\}$.

Figure 2 shows an example paper recommendation problem in which it will sometimes be more informative to query the noisy Professor 2 over the more rational Professor 3. This is because the frequency with which a noisy teacher prefers a lower-reward item over a higher-reward one gives information about the difference between the rewards, and in this example the recommender must learn *how much* more relevant Application papers are than Benchmark papers. Without this information, the system cannot distinguish between cases where $\mathcal{U}(A) = 8$ (indicating that the expected relevance of ICLR is greater than ICML) and where $\mathcal{U}(A) = 6$ (indicating the reverse).

**Experiments** We evaluate all algorithms for 25 runs of 1000 timesteps on 20 different paper recommendation tasks. Each task is a HUB with $\mathcal{I}$, $\mathcal{C}$, and $\beta$ as described above and a unique tuple $\langle \mathcal{U}, \mathcal{D}^{\mathcal{C}} \rangle$. $\mathbb{U}$ and $\mathbb{D}$ are discretized, and each task's $\langle \mathcal{U}, \mathcal{D}^{\mathcal{C}} \rangle$ is chosen such that $c^1$ has the highest

---

[3]Example relevance scores and paper category compositions were selected arbitrarily.

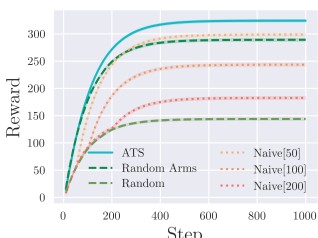
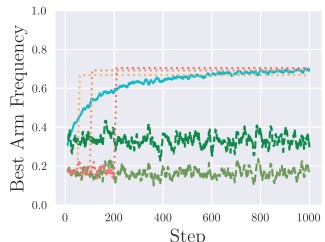
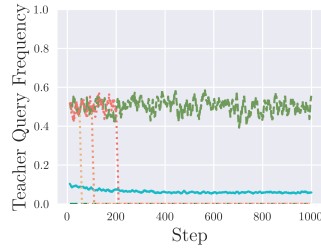

(a) Discounted cumulative reward  (b) Frequency of pulling best arm  (c) Frequency of querying teacher

Figure 3: Comparison of ATS, naive and random algorithms. ATS best maximizes discounted reward (a) and identifies the highest-reward arm more often than most baselines and comparably with Naive[100] and Naive[200], which explore more and earn less reward (b). ATS initially queries teachers less often than naive baselines, but continues querying teachers throughout the episode (c). All data is averaged across 25 runs on 20 HUB problems and smoothed over 10 steps. The legend applies to all plots.

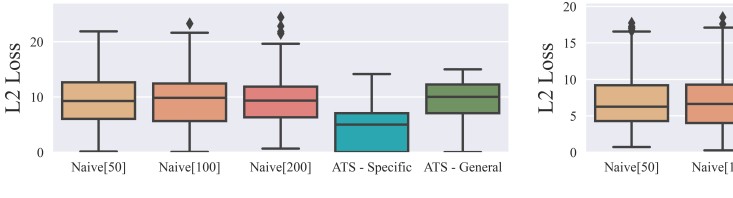

(a) $L2$ loss of $\mathcal{U}$ estimate  (b) $L2$ loss of arm reward estimate

Figure 4: Accuracy of reward learning using ATS (with specific and general teacher selection) and naive algorithms (with exploration parameters of 50, 100, and 200). ATS with specific teacher selection learns both the underlying utility function (a) and the expected rewards of each arm (b) much more accurately than ATS with general teacher selection and naive algorithms. Accuracy is measured as L2 loss, aggregated across 25 runs on 20 HUB problems. The middle line is the median, the boxes are the IQR, the whiskers are $1.5$ times the IQR, and the diamonds are outliers.

expected relevance ($\mathbb{E}[\mathcal{U}(i \sim c^1)] > \mathbb{E}[\mathcal{U}(i \sim c^2)] \geq \mathbb{E}[\mathcal{U}(i \sim c^3)]$) and all paper distributions are different and non-deterministic ($\mathcal{D}^j \neq \mathcal{D}^k \; \forall_{j,k \in \mathcal{C}}$ and $\mathcal{D}^c(i) \neq 1.0 \; \forall_{i \in \mathcal{I}, c \in \mathcal{C}}$).

**Results** While all non-random algorithms successfully identify the most relevant conference in expectation (Figure 3b), ATS with specific teacher selection best balances querying teachers with recommending papers, achieving the highest average discounted cumulative reward (Figure 3a), and most accurately learning relevance scores (Figure 4).

Figure 3b shows how often each algorithm learns to pull the best HUB arm and therefore recommend the most relevant conference over the course of training. All HUB solution methods (ATS, Naive[50], Naive[100], Naive[200]) successfully identify the most relevant conference, recommending it about three times as often as they would if they were behaving randomly ("Random" baseline, light green line) and about twice as often as if they were blindly recommending conferences ("Random Arms" baseline, dark green line). This indicates that the HUB formalism can be used to accurately represent the paper recommendation problem.

While all solution methods identify the best arm, ATS does so most efficiently, querying teachers sparingly even at the start of the task (Figure 3c) and best optimizing the HUB objective of expected discounted cumulative reward (Figure 3a). Moreover, ATS forms the most accurate estimates of the utility function and expected conference relevance scores (Figure 4) after 1000 timesteps, while continuing to explore and potentially improve this estimate by occasionally querying teachers and recommending other conferences (Figure 5a). In contrast, Naive algorithms stop learning after their hand-specified exploration horizon (Figure 5b), and Random algorithms never learn at all (Figure 5c). This demonstrates the benefits of actively selecting when to query teachers, as in ATS, rather than following a predefined schedule, as in standard RLHF.

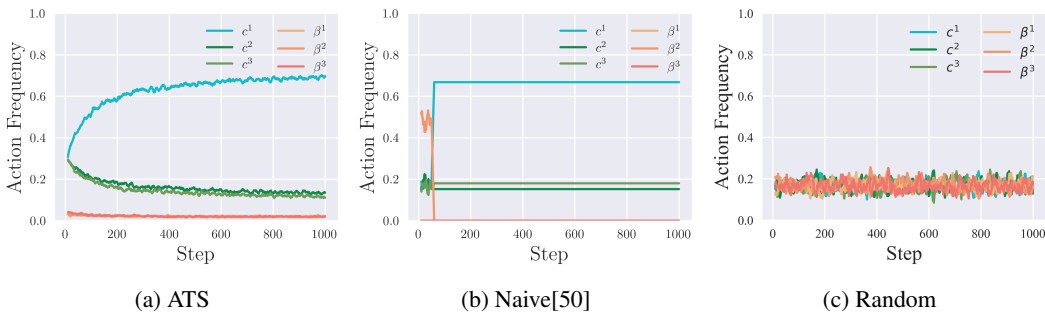

Figure 5: Mean action frequencies for various algorithms. $c$ actions are arm pulls and $\beta$ actions are teacher queries. Data is averaged across 25 runs of 20 HUB problems and smoothed over 10 steps.

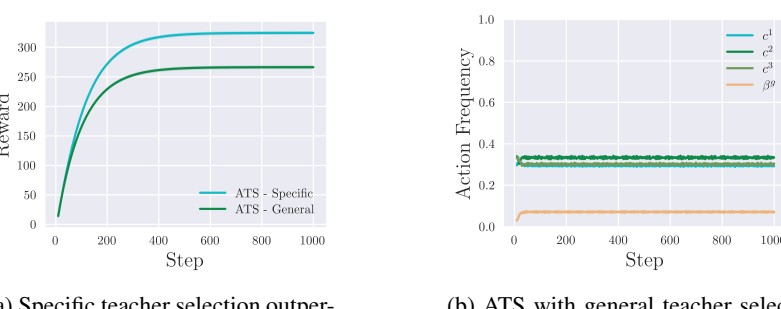

(a) Specific teacher selection outperforms general teacher selection.

(b) ATS with general teacher selection doesn't identify the best arm.

Figure 6: Performance of ATS with specific and general teacher selection. All data is averaged across 25 runs on 20 HUB problems, smoothed over 10 steps, and discounted with $\gamma = 0.99$.

Figure 6 compares ATS with specific and general teacher selection. Standard RLHF systems do not allow the agent to select which teacher to query and are most akin to general selection. However, we show that the additional control afforded by specific selection allows ATS to make more informative queries. Figure 6a shows that ATS with specific teacher selection earns higher expected reward than ATS with general teacher selection, and Figure 6b shows that ATS with general teacher selection queries all arms roughly equally, failing to identify the one with highest expected reward.

## 5.2 COVID-19 VACCINE TESTING DOMAIN

Bandit-type problems are commonly used to model medical treatment investigation, so as a proof-of-concept we apply the HUB framework to a real-world medical problem: evaluating vaccines for the 2019 Novel Coronavirus (COVID-19). This task is complicated by the difficulty of evaluating whether a patient is infected: many infections are asymptomatic, and other common illnesses cause similar symptoms. There are a variety of ways to test whether patients have COVID-19, including symptom surveys, antigen tests, and RT-PCR tests, but these vary widely in accuracy and cost.

The HUB framework directly models these challenges. Let the item set be easily observable patient symptoms, $\mathcal{I} = \{\text{None}, \text{Cough}, \text{Fever}\}$. The "arms" are vaccine candidates, $\mathcal{C} = \{c^1 = \text{VaccineA}, c^2 = \text{VaccineB}, c^3 = \text{NoVaccine}\}$, and the "teachers" are COVID-19 test types,

| Symptoms | Utility |
|----------|---------|
| None | 8.0 |
| Cough | 3.0 |
| Fever | 0.5 |

| Test | Rationality | Cost |
|------|-------------|------|
| Survey | 0.36 | -0.006 |
| Antigen | 1.32 | -0.21 |
| RT-PCR | 2.54 | -0.31 |

| Vaccine | Vaccine A | | | Vaccine B | | | No Vaccine | | |
|---------|-----------|---|---|-----------|---|---|------------|---|---|
| Symptom Distribution | N | C | F | N | C | F | N | C | F |
| | 0.9 | 0.1 | 0 | 0.6 | 0.3 | 0.1 | 0.5 | 0.3 | 0.2 |
| Expected Utility | 7.5 | | | 5.75 | | | 5.0 | | |

Figure 7: COVID-19 vaccine testing as a HUB problem. Symptoms (None, Cough, Fever) are items ($\mathcal{I}$), tests are teachers with rationality ($\beta$) and cost ($F$) parameters, and vaccines are arms ($\mathcal{C}$) with the specified distributions over patient symptoms ($\mathcal{D}$).

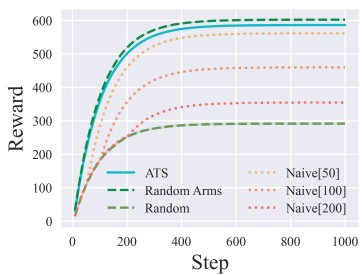 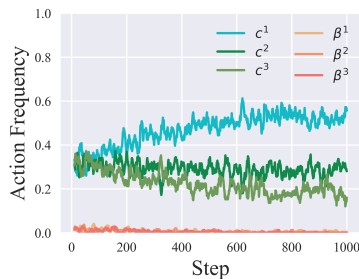

(a) Discounted cumulative reward        (b) ATS action frequencies

Figure 8: Performance of all algorithms and ATS action frequencies on the COVID-19 vaccine testing problem. Random Arms and ATS both earn high reward from frequently vaccinating participants (a), though only ATS additionally identifies the most effective vaccine (b). Data is averaged across 25 runs and smoothed across 10 steps.

$\{\text{Survey}, \text{Antigen}, \text{RT-PCR}\}$. Surveys are the least accurate but least expensive, while RT-PCR tests are the most accurate and most expensive. We estimate the US dollar cost of surveys at \$1.20 (accounting for 10 minutes of time at the US federal minimum wage of \$7.25), antigen tests at \$42, and RT-PCR tests at \$62 (median prices reported by (Lo et al., 2023)), then scale these costs by 0.05. We estimate $\beta$ by gathering real-world data on the sensitivity of COVID-19 symptom surveys (Rufino et al., 2023), antigen tests (Harmon et al., 2021), and RT-PCR tests (Binny et al., 2023), interpret this sensitivity as the probability $P$ of the test "preferring" a patient with no COVID-19 ($\mathcal{U} = u_{max}$) to a patient with definite COVID-19 ($\mathcal{U} = u_{min}$), let $\Delta_{ij} = u_{min} - u_{max}$, and calculate $\beta_m$ using Equation 3. We construct arm distributions where patients display the most frequent and severe symptoms with no vaccination, and the least symptoms with Vaccine A, and a utility function where symptoms that have a greater chance of indicating COVID-19 infection have lower scores. These values are reported in Figure 7.

**Experiments** We evaluate all algorithms for 25 runs of 1000 timesteps on this COVID-19 task. $\mathbb{U}$ and $\mathbb{D}$ are more finely discretized than in the recommendation HUB in order to allow for more realistic values, so the resulting HUB-POMDP has 5 times more states and is more challenging to solve. While the recommendation experiments averaged results over many problem parameters, here we fix the parameters to the values reported in Figure 7, since they are derived from real-world data and realistic estimates.

**Results** Figure 8 summarises the results. Several algorithms perform well: ATS, Random Arms, and Naive[50] (Figure 8a). The Random Arms baseline that randomly administers vaccines without testing for COVID-19 performs surprisingly well due to the high cost of reliable testing. However, in this domain, we care not only about vaccinating as many people as possible during the trial, but also about identifying which vaccine performs best. ATS clearly identifies the best vaccine, using it increasingly frequently during the trial (Figure 8b). The Naive algorithms also identify the best vaccine, but conduct more costly tests than necessary, leading to poorer overall performance.

## 6    CONCLUSION

We formalized the teacher selection problem in reward learning and proposed a solution method that expresses this problem as a POMDP. Our empirical results underscore the applicability of this framework to real-world problems, as well as the importance of modeling human teachers as distinct entities and actively choosing *when* and *which* teacher to query.

The purpose of this paper is to investigate the novel problem of selecting teachers in RLHF, so the experimental domains focus on tasks where learning the utility function is more challenging than optimizing it. However, real-world RLHF systems often use large deep learning models to tackle challenging problems. Future work will scale insights gained from working with the HUB formalism to reward modeling for large language modeling and deep reinforcement learning systems.

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

## A    RELATED WORK

**Inverse Reinforcement Learning**  *Inverse reinforcement learning* (IRL) is a reward learning technique in which the agent infers a reward function given behavioral samples from an optimal policy (Ng & Russell, 2000; Abbeel & Ng, 2004) or a noisy teacher (Ziebart, 2010). It is similar to RLHF in that reward information comes from a teacher rather than the environment, but distinct in that it requires teachers to perform the task well themselves (Milli & Dragan, 2020). RLHF and the HUB framework are most useful in domains such as those presented in Section 5, where the teacher can distinguish good performance, but does not know how to produce it themselves.

**Cooperative Inverse Reinforcement Learning** *Cooperative inverse reinforcement learning* (CIRL) extends the IRL framework to allow cooperation and collaboration between the agent and the teacher (Hadfield-Menell et al., 2016; Malik et al., 2018). HUB problems can be viewed as a specific class of CIRL games in which there are multiple humans, but they can only act (by providing feedback) when the agent requests it (by querying them). However, CIRL problems are DEC-POMDPS, which are NEXP-complete and thus functionally intractable (Bernstein et al., 2002). By fixing the human policy and arm distributions, the HUB framework reduces the problem to a POMDP with a stationary transition function, which is much more tractable. Optimal agent solutions to the CIRL game balance inference and control to produce several qualitatively valuable behaviors, such as only asking the human questions when necessary (Shah et al., 2020). The algorithm that best solves the HUB problem, ATS, demonstrates similarly conservative querying behavior.

**Crowdsourcing**  Prior work has investigated the related problem of combining feedback from multiple noisy annotators (Dawid & Skene, 1979), often to label training data for supervised learning algorithms. (Raykar et al., 2010) present an approach that learns teacher expertise and uses teacher feedback to fit a classifier simultaneously, while (Rodrigues et al., 2014) generalise gaussian process classification to model noisy annotators and combine their feedback into reliable labels for supervised learning. (Murugesan & Carbonell, 2017) develop a method that also models cost, trading off between querying noisy peer labelers and querying a costly oracle. This body of work underscores the difficulty and importance of combining feedback from varying and noisy teachers in machine learning.

## B    THEOREM 1 PROOF

**Theorem 1.** *If the predicted utility function $\hat{\mathcal{U}}$ and the predicted arm distribution $\hat{\mathcal{D}}^{\mathcal{C}}$ are estimated by executing Algorithm 1 with $T$ samples, then $\hat{\mathcal{U}} \to \mathcal{U}^*$ and $\hat{\mathcal{D}}^{\mathcal{C}} \to \mathcal{D}^{\mathcal{C}*}$ as $T \to \infty$.*

*Proof (Sketch).*  Since the number of arms is finite and they are pulled uniformly as $T \to \infty$, the number of times that a given arm $c^k$ is pulled approaches infinity. Since each pull samples an item from the true distribution $\mathcal{D}^{k*}$ i.i.d., the empirical distribution $\hat{\mathcal{D}}^k$ will approach $\mathcal{D}^{k*}$ in the limit of infinite pulls. This argument applies for all arms $c^k \in \mathcal{C}$, so $\hat{\mathcal{D}}^{\mathcal{C}} \to \mathcal{D}^{\mathcal{C}*}$ as $T \to \infty$. Similarly, in the limit of infinite queries, $\hat{P}(\beta, (i, j))$ will approach $P^*(\beta, (i, j)) = \Pr(i \succ j; \beta, \mathcal{U}^*)$, the true probability that teacher $b$ prefers item $i$ over item $j$, as determined by Equation 1. Given $\beta, (i, j)$ and $\hat{P}(\beta, (i, j))$ from the first $T$ timesteps, we can calculate $\Delta_{ij} = \hat{\mathcal{U}}(i) - \hat{\mathcal{U}}(j)$ using Equation **??**. Given $\Delta = [\Delta_{01}, \Delta_{02}, \ldots, \Delta_{NN}]$, $u_{max}$ and $u_{min}$, we can calculate $\hat{\mathcal{U}}$ as described in Algorithm 1. $\hat{\mathcal{U}} \to \mathcal{U}^*$ as $\hat{P} \to P^*$, which occurs as $T \to \infty$.  □

## C    THEOREM 2 PROOF

**Theorem 2.** *Given two items $i, j \in \mathcal{I}$ where $\mathcal{U}(i) < \mathcal{U}(j)$ and the preference probability $P = \Pr(i \succ j; \beta_m, \mathcal{U})$ from Equation 1 we can estimate $\hat{\beta}_m = \frac{1}{z}\beta_m$ as in Equation 3. If $\Delta_{ij}$ is known, we can further calculate $\beta_m = z \cdot \hat{\beta}_m$, where $z = -\Delta_{ij}^{-1}$.*

$$\hat{\beta}_m = \ln\left(\frac{1}{P} - 1\right). \tag{3}$$

*Proof (Sketch).* First, we define an affine mapping function $f_{a,b}(x) = ax + b$ such that $f_{a,b}(\mathcal{U}(i)) = 0$ and $f_{a,b}(\mathcal{U}(j)) = 1$. Lemma 3 shows that this is always possible when $\mathcal{U}(i) \neq \mathcal{U}(j)$ and furthermore that $a = \frac{-1}{i-j}$. Let $z$, $y$ be the parameters that make this mapping for these particular values of $\mathcal{U}(i)$ and $\mathcal{U}(j)$. Note that $z = \frac{-1}{i-j} = -\Delta_{ij}^{-1}$.

Next, suppose we have that $\beta'_m = \frac{1}{a}\beta_m$, it follows that:

$$
\begin{aligned}
P &= \Pr(i^0 \succ i^1; \beta_m, \mathcal{U}) \\
&= \frac{\exp(\beta_m \mathcal{U}(i))}{\exp(\beta_m \mathcal{U}(i)) + \exp(\beta_m \mathcal{U}(j))} && \text{(by Equation 1)} \\
&= \frac{\exp(\frac{\beta_m}{a} \cdot a\mathcal{U}(i) + \frac{\beta_m}{a}b)}{\exp(\frac{\beta_m}{a} \cdot a\mathcal{U}(i) + \frac{\beta_m}{a}b) + \exp(\frac{\beta_m}{a} \cdot a\mathcal{U}(j) + \frac{\beta_m}{a}b)} \\
&= \frac{\exp(\beta'_m \cdot (a\mathcal{U}(i) + b))}{\exp(\beta'_m \cdot (a\mathcal{U}(i) + b)) + \exp(\beta'_m \cdot (a\mathcal{U}(j) + b))} && \text{(by definition of } \beta'_m) \\
&= \frac{\exp(\beta'_m \cdot f_{a,b}(\mathcal{U}(i)))}{\exp(\beta'_m \cdot f_{a,b}(\mathcal{U}(i))) + \exp(\beta'_m \cdot f_{a,b}(\mathcal{U}(j)))} && \text{(by definition of } f_{a,b}) \\
&= \frac{\exp(0)}{\exp(0) + \exp(\beta'_m)} = \frac{1}{1 + \exp(\beta'_m)}.
\end{aligned}
$$

Finally, solving for $\beta'_m$ yields $\beta'_m = \frac{1}{z}\beta_m = \ln(\frac{1}{P} - 1) \quad \rightarrow \quad \beta_m = z \cdot \ln(\frac{1}{P} - 1)$. $\qquad\square$

**Lemma 3.** *Given any two numbers $m$, $n \in \mathbb{R}$ such that $m \neq n$, there exists an affine transformation $f_{a,b} : \mathbb{R} \to \mathbb{R}$ that maps the greater number to 1 and the lesser number to 0.*

*Proof (Sketch).* Suppose that $m > n$ without loss of generality. We therefore must solve the following system of equations: $f_{a,b}(m) = am + b = 1$ and $f_{a,b}(n) = an + b = 0$. The solution is $a = \frac{-1}{n-m}$ and $b = \frac{m}{n-m} + 1$, which always exists when $m \neq n$. $\qquad\square$

## D  POMCPOW ROLLOUT POLICIES

ATS solves the HUB-POMDP using *partially observable Monte-Carlo planning with observation widening* (POMCPOW) augmented with a custom rollout policy for estimating the value of leaf nodes in the search tree. We evaluate a *random action* rollout policy, which takes actions uniformly at random from $\mathcal{A} = \mathcal{C} \cup \beta$, a *random arm* rollout policy, which chooses arms uniformly at random from $\mathcal{C}$, and a *best arm* policy, which calculates which arm has the highest expected utility *according to the current belief $b$*, then always chooses that arm.

Since a utility-maximizing agent will choose arms more often if it believes them to have higher utility, the *best arm* policy rollouts most closely resemble the actions the actual policy would take from belief $b$, yielding the most accurate value estimates. As a result, ATS with best arm rollouts outperforms the alternatives on the paper recommender domain, as shown in Figure 9. Results are averaged across 25 runs on 20 different paper recommendation tasks.

## E  HUB COST EFFECTS

We investigate the impacts of teacher query cost on ATS performance by varying professor feedback costs in the paper recommendation domain. We set linear costs $F = \{-1, -2, -3\}$ and scale them by a *cost multiplier*. As in the other paper recommendation experiments, results are averaged across 25 runs on 20 different paper recommendation tasks.

We find that ATS responds rationally to changes in costs, querying teachers more sparingly (Figure 10b) and consequently identifying the best arm more slowly (Figure 10a as overall costs increase. This leads to a slight decrease in overall performance (Figure 10c).

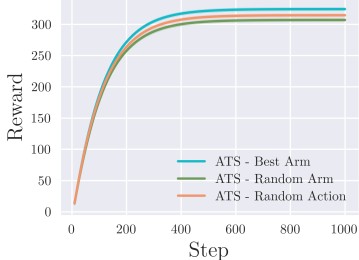

Figure 9: Performance of ATS with various rollout policies. The best arm rollout policy outperforms the random arm and random action rollout policies. All data is averaged across 25 runs on each of 20 HUB problems, smoothed over 10 steps, and discounted with $\gamma = 0.99$.

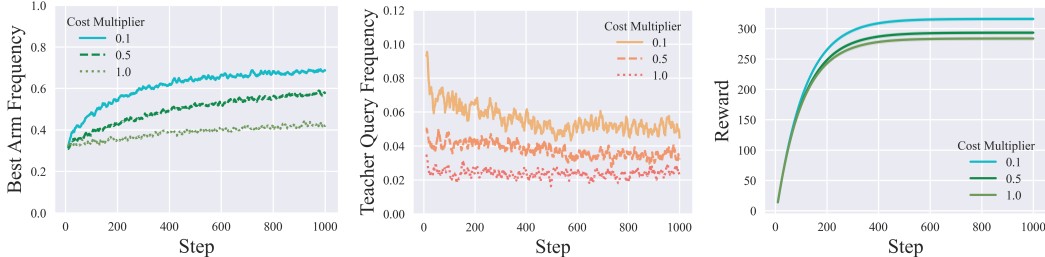

(a) Frequency of pulling the best arm    (b) Frequency of teacher queries    (c) Discounted cumulative reward

Figure 10: ATS behavior and performance varies with teacher query costs. Data is averaged across 25 runs on 20 paper recommendation HUB problems and smoothed over 10 steps.

