# OpenReview forum: "Active Teacher Selection for Reinforcement Learning from Human Feedback"
_ICLR.cc/2024/Conference — Submitted to ICLR 2024_

### Official Review · Reviewer_eqU7 · 2023-10-27

**Soundness:** 3 good
**Presentation:** 3 good
**Contribution:** 2 fair
**Rating:** 5
**Confidence:** 4

**Summary:**

This paper proposes an approach for selecting the best teacher (annotator) during the process of learning the reward function in a bandit-like setting. The characteristic feature of the setup is that several teachers are available with varying reliability but the same preferences. The authors introduce the problem of Hidden Utility Bandit where there is a choice between drawing an arm and querying the teacher about the utilities of the observations. The authors conduct experimental studies on paper recommendations application and on COVID-19 vaccine testing.

**Strengths:**

The paper is well written and easy to follow. The goal of the paper is to model the reward learning setup with more realism and to remove the assumption of having a single teacher and it is an important goal. I find the observation about the fact that not the most reliable teacher would be selected all the time (because the odds of the selection from less reliable teachers allow us to estimate the difference in utilities) quite interesting and insightful.

**Weaknesses:**

My main concern is about the applicability of the assumptions in the problem formulation to the real domains. When introducing HUB formalization, it would be useful to draw parallels with the real applications. This becomes more clear in the experimental studies, but then it seems to be a bit artificial. For example, it seems that there is an assumption that all the teachers have the same utility function (it would be useful if the paper stated all the assumptions of the introduced framework clearly, another one that is not stated clearly is that the space in which the reward is learnt is assumed to be discrete). I am not sure this assumption is suitable for the paper recommendation applications (section 4.1) as it means that everyone has the same preferences. Other examples: it seems that after asking for a preference in recommendation, one needs to wait for a day to propose an article, or the results of vaccine application are observed immediately etc. The studied setting sounds interesting, but so far I am not convinced that problem formulation fits the real applications well. I think in order to be more convincing, it might be useful to find another realistic application for the proposed method, or modify the assumptions of the method to fit the real world applications better.

I think the related literature lacks the literature on selecting the annotators when annotating data for supervised learning in crowdsourcing and active learning (see some examples, but more can be easily found).

F. Rodrigues, F. Pereira, and B. Ribeiro. Gaussian process classification and
active learning with multiple annotators.ICML, 2014.

C. Long, G. Hua, and A. Kapoor. Active visual recognition with expertise estimation
in crowdsourcing. In International Conference on Computer Vision, 2013

K. Murugesan and J. Carbonell. Active learning from peers. In Advances in Neural
Information Processing Systems, 2017

[BOOK] Human-in-the-Loop Machine Learning: Active learning and annotation for human-centered AI 2021

The question of the related literature brings me to the next point. The authors claim that this new problem formulation lacks the baselines in the related literature. However, as the problem has a component of selecting the annotators for crowdsourcing annotation and bandit problem that maximizes the reward, a simple baseline could be to do it in two stages: 1) active teacher selection to learn the reward function, 2) bandit algorithm to maximize the sum of rewards. Would such a baseline be applicable?

After reading the paper, I still have some questions related to the main method. What is the advantage of re-formulating the problem as POMDP (as it still has a single state as a bandit)? How does it relate to the contextual bandit algorithms? The NaiveHUBInference is described in many details, but the proposed method of the paper is very brief and lacks details. I would like to hear more about it. Is the policy learnt on the fly or some prior simulations are needed?

Some minor questions:
- for estimating \beta the authors propose to use a dataset of preferences, however, if such a dataset is available, the reward might be learnt directly and used in the future.
- Information in tables could be made a bit more readable (Figure 2 and 7)
- RLHF term in the title is confusing. The focus of the paper is on rewards learning, and it could be better reflected in the title.

**Questions:**

I would like to hear from the authors regarding my concerns on the applicability of the method assumptions to real problem domains, how the method related to the annotator selection in crowdsourcing literature and if comparison to such a method is applicable.

---

> ### Author Response · Authors · 2023-11-22
>
> Thank you for the detailed feedback and suggestions. We found the result that querying noisier teachers can sometimes be more informative to be surprising ourselves, and are pleased that you found it insightful.
>
> We particularly appreciate your suggestions of ways to improve the paper. In response, we’ve made the following revisions:
> * We state assumptions more explicitly. When presenting the HUB problem, we state the assumptions that all actions take one timestep (which we agree is a simplification, but a standard one in sequential decision-making literature) and that all teachers reference the same utility function (which is also a simplification, but a necessary one as optimal preference aggregation is an unsolved problem and solving it is out-of-scope for this work). (Section 3, page 3)
> * We add additional details to the simple HUB problem presented at the start of the paper to make it more concrete. (Section 1, page 1)
> * We add a discussion of crowdsourcing including the proposed references and a few additional ones. (Appendix A, page 13)
>
> In response to your questions about the method, we used a POMDP rather than contextual bandit framing because the contextual bandit doesn’t apply to our problem. In the HUB problem, the state (utility function, arm distributions over items, teacher parameters) never change, so there is no changing “context” to model. ATS maintains and updates a belief over the utility function and arm distributions, so this changing belief functions as a changing state. However, the belief changes in response to actions (for example, querying a teacher can update the belief over the utility function), violating the contextual bandit assumption that state changes are independent of actions. POMDPs are designed for this scenario, so we use a POMDP formulation and solve it with adapted POMCPOW. As an online Monte Carlo planner, POMCPOW generates simulations and determines the policy online (“on-the-fly”).
>
> In response to your question about estimating utilities, we’d like to clarify that preferences depend on both the utility function and the teacher rationality parameter $\beta$ as shown in Equation 1. As a result, utilities cannot be calculated from preferences without also knowing $\beta$. Prior RLHF research typically makes the simplifying assumption that $\beta$ is known, allowing utilities to be calculated from preferences. However, we believe that this is not always a realistic assumption, so we also present a method to estimate $\beta$ (Section 4.2) and demonstrate the procedure in a real-world domain (Section 5.2).
>
> Thank you also for suggesting an additional baseline. We believe that this baseline would be applicable, but would perform significantly worse than our existing Naive baseline (while being more complicated because it relies on ATS and a bandit algorithm). If we understand correctly, your proposed stage 1 would be comparable to a lower-reward version of the exploration period in the Naive algorithm. ATS selects which teacher to query based on (1) teacher query cost and (2) expected usefulness of teacher feedback for identifying higher-utility arms. (2) requires a belief over arm distributions, which ATS must pull arms in order to update. Therefore, if ATS only queries teachers in stage 1, it can only select teachers based on (1), which will result in it always querying the lowest-cost teacher (or choosing randomly, if there isn’t a unique lowest cost). This is comparable to the Naive algorithm always querying the same teacher during its exploration period. However, the Naive algorithm intersperses teacher queries (which impose a cost) with arm pulls (which earn a utility). Since the HUB problem is temporally discounted, it is better to intersperse positive and negative reward actions early on, rather than doing a block of negative reward ones first. Moreover, in the worst case the lowest-cost teacher will be essentially random, and teacher-only ATS won’t learn anything. Your proposed stage 2 (using a bandit algorithm) would also be more complicated than the post-exploration phase of the Naive algorithm (which is deterministic). If stage 1 learns a good utility function, the bandit algorithm may do a better job than the Naive one at finding the highest-reward arm. However, the Naive algorithm identifies the highest-reward arm quite frequently already, so it’s unlikely that this would offset the lost reward from front-loading teacher query costs.

---

### Official Review · Reviewer_9GWq · 2023-10-29

**Soundness:** 2 fair
**Presentation:** 2 fair
**Contribution:** 2 fair
**Rating:** 5
**Confidence:** 4

**Summary:**

The authors study the problem of reinforcement learning from a set of distinct teachers instead of a single teacher. To solve this problem, the authors develop a hidden utility bandit framework to model the teacher’s rationality and querying cost.  Specifically, this paper proposed an active teacher selection algorithm to decide when and which teacher to choose. Experiments are carried out on the paper recommendation task and the COVID-19 vaccine testing task.

**Strengths:**

1.	The studied problem is timely and interesting. Current RLHF assumes all the human feedback is collected from a single human teacher instead of a variety of teachers, which is the focus of this paper.
2.	The authors formulate the teacher selection problem in a hidden utility bandit (HUB) framework. Naïve MAB algorithm does not decide when and which teacher to select. The proposed active teacher selection algorithm solves the hub problem by maintaining the belief over the utility function to decide the time and which teacher to query.
3.	Experiments on the paper recommendation and Covid-19 vaccine testing problem demonstrate the effectiveness of the proposed problem.

**Weaknesses:**

1.	The authors claim that the naïve bandit algorithm does not decide when to query the teachers. However, one simple modification of the naïve bandit algorithm is to add a binary action to decide whether to query the teacher based on the state by extending it to the contextual bandit setting. The authors need to justify the strengths of the proposed algorithm over this simple algorithm.
2.	There is no theoretical guarantee of the regret of the proposed algorithm.
3.	In the experiments, it’s better to demonstrate the number of query times to show the efficiency of the learning by querying teachers.

**Questions:**

See the Weaknesses for the questions.

---

> ### Author Response · Authors · 2023-11-22
>
> Thank you for your review. We appreciate that you found the problem we present to be timely and interesting.
>
> While your suggested baseline is interesting, it is extremely similar to one of the variants of the algorithm that we present (ATS). In the ATS-general variant, we add a binary action to decide whether to query a teacher, as you suggest. We agree that this is an important baseline for comparison, since standard RLHF systems similarly do not distinguish between teachers. We compare this to the ATS-specific variant, with a query action for each possible teacher, at the end of section 5.1 and Figure 6 (page 8). We find that ATS-specific performs significantly better than ATS-general.
>
> ATS-general uses a POMDP formulation rather than a contextual bandit one as you suggest because the contextual bandit setting is not equipped to handle a continuous high-dimensional state space that changes in response to actions taken. In our problem, the state space is a belief (continuous) over multiple distributions (high-dimensional) that update in response to actions (for example, updating utility estimates based on teacher feedback). This violates the contextual bandit assumption that pulling arms doesn’t impact state, so we must use POMDPs instead.
>
> We agree that it is informative to illustrate the amount of teacher queries. We do so in Figure 3c (showing how teacher queries vary across time and algorithm) and in Figure 10b (showing how teacher queries vary with teacher cost).

---

### Official Review · Reviewer_szYk · 2023-10-31

**Soundness:** 3 good
**Presentation:** 3 good
**Contribution:** 2 fair
**Rating:** 5
**Confidence:** 5

**Summary:**

In this work, the author study a hidden bandit model with an option obtain item comparison feedback from a set of expert teacher. There formulation require solving a POMDP which is intrinsically hard to solve in practice so they propose to use a off-the-shelf Monte Carlo planner to obtain an approximate solution to the POMDP problem. Further, they performed experiments on two real world examples of Conference Recommendation and COVID-19 Vaccine testing to demonstrate the effectiveness of their method.

**Strengths:**

- Proposes an interesting bandit model that combines direct feedback from arms and comparison feedback from experts.
- They conduct several empirical experiments to validate effectiveness of their algorithm.
- They discusses how to infer noise parameter when item utilities are known.

**Weaknesses:**

1. Notations and problem setup can be significantly improved. The problem clearly seems to be a simpler partially observable bandits problems with additional preference based feedback. But the authors have unnecessarily complicated it to be a POMDP and using a discounted cumulative value.
2. There is no discussion about tradeoff between reward obtained from actual arms and cost of querying the teacher. If cost of querying the expert is low, is it ok to just keep querying them to maximize the cumulative return even if does not help learning about the optimal arm?
3. The paper is easy to follow but has bunch of notation issues that can be fixed.
    a\. Rewards and costs have been mixed together.. The cost for teacher should be decreased while utility for arms should be increased. I guess the authors meant to use $1/f^a$ or -$f^a$ as corresponding reward function when querying teacher.
    b\. Typo : $f^{\beta_t}$ is ill defined.
    c\. Formal definition of terms like $\mathcal U^\*, \mathcal D^{\mathcal C\*}$  is missing.
    d\. Figure 3.b, 3.c lack legends.
4. No theoretical guarantee on algorithm is provided.

**Questions:**

1. In section 3.2 authors assume that $\Delta_{ij}$​’s or $\mathcal U$ is known. In that case, what is there to learn more? The learner can simply estimate the multinomial parameter for each arm and compute the optimal arm.
2. In real world both $\beta$ and $\mathcal U$ would be unknown and has to estimated. How do you intend to handle this?

---

> ### Author Response · Authors · 2023-11-22
>
> Thank you very much for your review. In response to your points:
>
> * We’d like to clarify that we adapted an existing algorithm (POMCPOW) to our HUB problem by designing specialized rollout policies, rather than using an off-the-shelf planner. We’ve clarified this in the paper (final paragraph of Section 4.1,  page 5). We further discuss the specific rollout policies that we designed and results from empirical evaluations in Appendix D.
> * The HUB problem is not technically a bandit problem because the agent cannot observe utility from pulling arms. We realize that some of our wording was ambiguous, and made minor edits throughout to make this more clear. Because the items and teacher feedback are observed but the utilities are not, this is a partially-observable problem, and therefore a framework for sequential decision-making in partially observable domains is needed. We use POMDPs because they are well-suited to this.
> * The utilities from querying arms and costs from querying teachers are both combined into the objective (discounted sum of reward), so the agent must trade off between them. We state this explicitly in Section 3 (page 3). We also investigate the impact of increasing teacher cost and how it trades off against performance and cumulative reward in Appendix E. In response to your specific question, querying a low-cost teacher has the opportunity cost of preventing the agent from pulling arms to earn utility, so continuously doing so won’t maximize cumulative reward.
> * In Section 4.2 (previously 3.2), we show how to estimate $\hat{\beta}$ when $\mathcal{U}$ is not known. For completeness, we also show how to calculate $\beta$ directly when $\Delta_{ij}$ is known (which is a more lax requirement than knowing $\mathcal{U}$). However, this isn’t necessary. We’ve revised the wording in Section 4.2 (page 5) and Theorem 2 to make this more explicit.
>
> Thank you also for pointing out confusing notation. In response we made these minor revisions:
> * We changed the teacher costs from negative to positive values so that we can subtract them from the reward for the HUB-POMDP as you suggested. (Definition 4.1, page 5)
> * We clarified that $\mathcal{U}^*$ and $\mathcal{D}^C*$ are the ground truth values (Section 3.1, page 4).
> * We clarified that the legend applies to the full figure for Figure 3. (page 7)

---

### Official Review · Reviewer_WRYj · 2023-11-02

**Soundness:** 3 good
**Presentation:** 3 good
**Contribution:** 3 good
**Rating:** 6
**Confidence:** 3

**Summary:**

The paper considers active teacher selection within the realm of RLHF,  tackling pivotal queries pertaining to the optimal teacher to consult and the most opportune moments for such consultations. Many existing works assumes the feedback comes from a single human teacher. The author(s) proposed a hidden utility bandit framework to mathematically formulate the problem, modelled the data generating process via POMDPs and developed active teacher selection algorithms for regret minimisation.

**Strengths:**

**Originality**: The manuscript tackles a novel and underexplored issue within the context of RLHF. To the best of my knowledge, there exists a limited number of theoretical and methodological papers on this subject. Many existing work considers scenarios featuring a single human instructor responsible for delivering feedback. The active teacher selection has not been explored comprehensively in the literature, rendering this paper a valuable contribution to the field.

**Clarity**: The writing is generally clear.

**Significance**: The issue of active teacher selection holds considerable weight in practical scenarios, particularly given the potential high costs associated with soliciting input from teachers. The paper sheds light on intriguing findings that are in concordance with established knowledge in the field. For instance, it highlights instances wherein seeking insights from less knowledgeable instructors could yield richer information, thereby enhancing the accuracy of model parameter estimation.

**Quality**: The paper stands out for its soundness, evident in the problem formulation, the underlying principles of the proposed solution, and the methodological approach adopted. The authors have demonstrated proficiency in laying down a solid mathematical foundation, which underpins the validity and robustness of their work.

**Weaknesses:**

While the paper makes significant strides in addressing the active teacher selection problem within RLHF, certain aspects could benefit from further refinement and elaboration.

**Theoretical Justification**: The theoretical underpinnings of the algorithms presented in the manuscript are somewhat lacking. The paper does present two theories, yet these appear to be more of preliminary lemmas than comprehensive theoretical validations. A better analysis would entail a comparative study of the regret associated with the proposed algorithm against established benchmark methods, such as consistently relying on a single teacher. This comparative analysis would provide readers with a clearer understanding of the advantages and potential drawbacks of the proposed method.

**Clarity and Notation**: The manuscript exhibits areas of ambiguity, marked by several instances of unclear notation and expressions, particularly in Sections 2 and 3. The abundance of notation can be overwhelming, detracting from the overall readability of the paper. A revision aimed at simplifying and clarifying these sections would enhance the manuscript's accessibility. Additionally, the HUB framework, a central component of the paper, is not adequately explained. The manuscript would greatly benefit from the inclusion of detailed examples illustrating the practical application of this framework. Providing clear, real-world scenarios that elucidate the concepts of 'arms' and 'items' within the context of the HUB framework would render the paper more informative and user-friendly.

**POMDP Literature**: Given that the paper addresses the problem using POMDPs, it is imperative to include a thorough discussion of relevant POMDP literature. Incorporating references and discussions around works such as https://www.cs.cmu.edu/~ggordon/boots-siddiqi-gordon-closing-loop-psrs.pdf, https://arxiv.org/abs/2006.12484, https://arxiv.org/pdf/2207.13081.pdf, alongside an exploration of their relevance to the current study, would lend greater depth and context to the paper.

**Questions:**

* Page 3, shall $\mathbb{D}=\mathbb{D}^1\times \mathbb{D}^2\times \cdots \times \mathbb{D}^K$ be $\mathcal{D}=\mathcal{D}^1\times \mathcal{D}^2\times \cdots \times \mathcal{D}^K$?
* Page 3, shall the utility for teacher query cost be $-f^{\beta_t}$ instead of $f^{\beta_t}$?
* There is a question mark at the bottom of Page 3.
* What are the arms in practice?

---

> ### Author Response · Authors · 2023-11-22
>
> Thank you for the detailed and helpful feedback. We’re pleased that you agreed with the significance of the teacher selection problem and found our mathematical foundation and methodology to be sound, and we appreciate the questions and suggestions on how to improve the paper’s clarity.
>
> In response to your feedback, we made the following revisions:
> * We added further explanation of “arms” and “items” to the introduction, and described a concrete example (in which arms are vending machines, items are fruit, and teachers are taste-testers). (Section 1, page 1)
> * We added a “Background” section, including a discussion of the POMDP literature. We didn’t find the suggested papers on predictive state representations, undercomplete POMDPs, or off-policy evaluation sufficiently related to our work to include, but did add references to core papers on POMDPs and online simulation-based solvers. (Section 2, page 2)
> * We intended for the teacher costs to be *negative*, so it’s correct to *add* them in Definition 4.1 (originally 3.1). However, we see that this was confusing, so we revised to assume the costs to be *positive*, and *subtract* them from the reward for the HUB-POMDP. (Definition 4.1, page 5)
> * We’ve fixed broken links to the appendices throughout.
>
> We also want to clarify the difference between $\mathbb{D}$ and $\mathcal{D}$. As defined in Definition 3.1 (originally 2.1), $\mathcal{D}$ refers to a specific joint distribution, whereas $\mathbb{D}$ refers to a *space* of joint distributions. We didn’t see a reference to $\mathbb{D}$ page 3 of the original submission, but we did see a reference to $\mathcal{D}^\mathcal{C}$. This notation is correct, as it refers to a specific joint arm distribution. Please let us know if we can clarify further.

---

### Author Response · Authors · 2023-11-22

We’d like to thank all of the reviewers for their helpful feedback. We appreciated comments that the problem we present is timely and novel, and that our methodology clearly demonstrates the effectiveness of the HUB problem and ATS solution.

We also particularly appreciated suggestions on notation to clarify, references to incorporate, and related areas to discuss. We list the specific revisions made in response to each review in the comments below. We believe that these revisions significantly clarify the paper, improving our notation and better situating HUB and ATS with respect to related work.

---

### Meta-Review · Area_Chair_i79V · 2023-12-13

**Metareview:**

### Summary
This paper introduces the Hidden Utility Bandit (HUB) framework to address the limitation of assuming a single human teacher in RLHF settings. The paper tackles the challenge of modeling differences in teacher rationality, expertise, and costliness when learning from multiple teachers. The proposed Active Teacher Selection (ATS) algorithm outperforms baseline methods by actively choosing which teacher to query and when, as demonstrated in real-world domains such as paper recommendation systems and COVID-19 vaccine testing. This paper argues the importance of leveraging teacher differences for accurate reward modeling, aiming to pave the way for future research in active teacher selection for robust learning.

### Decision

Overall, the paper is clear and easy to follow. It is studying an important problem of learning from multiple teachers using RLHF. The approach is novel, and this problem is largely understudied. However, as pointed out by the reviewers, the paper is not ready yet. There are some ambiguities in the paper that need to be clarified further to make sure that the paper meets the ICLR's bar. I recommend the paper consider incorporating the suggestions made by the reviewers into their paper and submitting it to a different venue. Here are some of the weaknesses pointed out by the reviewers:

**Theoretical Foundations**
- WRYj: Lack of comprehensive theoretical validation, needing comparative analysis against benchmark methods.
- eqU7: Questioning the applicability of assumptions to real domains, unclear connections to real-world scenarios.

**Clarity and Notation**
- WRYj: Ambiguity in notation, the overwhelming abundance of notation.
- szYk: Notation issues, confusion between rewards and costs.

**Missing Theoretical Guarantees**
- szYk & 9GWq: Lack of theoretical guarantees on algorithm performance.

**Problem Setup and Justification**
- szYk: Complexity of problem formulation (POMDP) versus simpler alternatives.
- eqU7: Unclear problem formulation's relevance to real applications.

**Insufficient Explanations and Details**
- eqU7: Insufficient details about the proposed method, lack of clarity on advantages of reformulating as a POMDP.

**Relation to Existing Literature**
- eqU7: Insufficient discussion on related literature (e.g., selecting annotators, bandit algorithms) and baselines.
- 9GWq: Lack of justification for the proposed algorithm's strengths over more straightforward modifications.

**Experimental Insight**
- 9GWq & eqU7: Lack of demonstration regarding efficiency (e.g., number of query times) in experiments.

**Justification For Why Not Higher Score:**

The paper seems to be rushed, and I would highly recommend the authors fix the notation and issues suggested by the reviewers and consider resubmitting to a different venue. These substantial changes in the paper imply that the paper would benefit from another iteration and re-review.

**Justification For Why Not Lower Score:**

N/A

---

### Decision · Program_Chairs · 2024-01-16

Reject